# Modeling Finance–Growth Nexus in MENA Region Economies: A Panel Data Analysis

Abdelmonem Lotfy Mohamed Kamal [1] , Mostafa E. AboElsoud [2,3,*] and Khaled Abdella [2,4]

1   Department of Finance and Investment, Faculty of Business Administration, Economics, and Information System, Misr University for Science and Technology (MUST), 6th of October City, Giza 11556, Egypt; abdelmonem.lotfy@must.edu.eg
2   Department of Economics, Faculty of Business Administration, Economics & Political Science, The British University in Egypt, Cairo 11837, Egypt; khaled.abdella@bue.edu.eg
3   Department of Economics, Faculty of Commerce, Suez Canal University, Ismailia 41522, Egypt
4   Department of Economics, Faculty of Management, Sadat Academy for Management Sciences, Cairo 11837, Egypt
*   Correspondence: mostafa.aboelsoud@bue.edu.eg

**Abstract:** The primary objective of this paper is to examine the relationship between finance and economic growth in a cohort of 16 economies within the MENA region spanning a four-decade period from 1980 to 2021. This study employs panel unit root and panel co-integration analyses to investigate this long-term nexus. The fully modified and dynamic Ordinary Least Squares (OLS) approaches are utilized to estimate the long-run coefficients. The findings underscore the existence of cross-sectional interdependence among these nations. Furthermore, Pedroni's panel co-integration research robustly supports the idea of a long-term co-integrating relationship between financial development and economic growth. Our long-run panel estimations reveal a positive and statistically significant impact of financial development on GDP per capita income growth. In addition to this broad analysis, this paper conducts a detailed time-series examination focused on a specific country to validate the robustness of the results. These findings further substantiate the favorable influence of financial development on income growth in the majority of MENA nations. Notably, private sector participation in these economies is found to be alarmingly low. As a result, a significant policy implication of this study underscores the urgent need for policymakers to prioritize measures conducive to private sector expansion. Moreover, enhancing financial inclusion, addressing the crowd-out effect, and tackling non-performing loans are critical areas requiring attention within the MENA region. Furthermore, our research highlights the potential benefits of developing stock markets as part of an optimal strategy to enhance both economic and income growth rates. In conclusion, this study contributes valuable insights into the finance–growth nexus in the MENA region, emphasizing the importance of financial development as a driver of economic prosperity and the need for targeted policy initiatives to support private sector growth and financial stability.

**Keywords:** financial development; income growth; finance–growth nexus; cross-sectional dependence; panel co-integration; MENA region economies





## 1. Introduction

A nation's economic growth, when accelerated, leads to the enhancement of the overall well-being of its population by amplifying productive capacity and advancing its fundamental infrastructure framework. Numerous scholars and policymakers have dedicated significant efforts to analyzing a diverse range of tools that foster economic growth. In recent years, there has been a growing emphasis on recognizing the pivotal role of the financial industry in contributing to overall economic growth.

The literature extensively acknowledges the significance of financial sector growth in facilitating economic progress, encompassing the development of both the stock market and

the banking industry. A robust financial system plays a crucial role in capital development and efficient resource allocation, promoting economic growth.

To comprehensively understand and thoroughly analyze the impact of the financial sector on economic growth, it is imperative to grasp the essential functions and significant contributions of this sector to the economy. Financial intermediaries are instrumental in facilitating technical innovation, economic growth, and development by overseeing managerial activities, mobilizing savings, managing risks, and facilitating transactions (Schumpeter and Opie 1934). According to Rajan and Zingales (1998), the financial market reduces borrowing costs, enabling organizations to expand their operations.

The investigation into the extent and direction of the relationship between financial development and economic growth holds utmost importance due to its potential to inform policy decisions that can significantly impact growth and improve living standards. Two perspectives emerge: the 'demand pull' theory posits that financial market expansion results from higher economic growth due to increased demand for financial services (Adu et al. 2013; Alhassan et al. 2022), while the supply-leading stance argues that economic growth follows financial development (Jalil and Feridun 2011; King and Levine 1993; Rajan and Zingales 1998; Sehrawat and Giri 2018). The premise that 'financial development matters for economic growth' was confirmed via comprehensive research conducted by the World Bank (1989) on developing nations implementing financial development programs. A robust financial market efficiently allocates financial resources to productive endeavors, fostering economic expansion, but opposing viewpoints also exist.

This paper aims to investigate the relationship between finance and economic growth in the Middle East and North Africa (MENA) region. It employs advanced quantitative approaches and the latest available data to address these challenges. This study contributes in three significant ways. Firstly, we utilize a log–log linear regression model to examine the relationship between finance and economic growth while considering the influence of other factors for reliable results. Second, despite the growing significance of the MENA region in the global economic landscape, research on the relationship between finance and economic growth in this region remains limited. To address this gap, we employ a comprehensive dataset spanning the years from 1980 to 2021, comprising panel data from 16 nations. Finally, we utilize the broad-based financial development index developed by Svirydzenka (2016) from the International Monetary Fund (IMF) to investigate the relationship between finance and economic growth in the MENA region, enhancing the robustness of our empirical results.

The structure of this paper is as follows: Following this introduction, Section 2 discusses and synthesizes a review of the relevant literature. Section 3 presents the models adopted in this paper and the data. The estimation techniques are discussed in Section 4. Empirical findings are presented in Section 5, followed by the conclusions in Section 6.

## 2. Literature Review

A substantial body of literature has delved into the significance of the relationship between financial markets and economic growth. It was Schumpeter and Opie (1934) who first recognized finance's central role in economic growth, igniting a new wave of scholarly discourse. Their argument posited that advancing economies stemmed from improved financial infrastructure, enabling greater capital accumulation and technological innovations. Since the groundbreaking work of King and Levine (1993), several research studies have explored the link between financial development and growth. However, discrepancies in the supporting data have persisted. Variations in sample sizes, timeframes, and quantitative methodologies may account for some of these discrepancies.

Nevertheless, a cadre of scholars have cast doubt on the purported growth-enhancing effects of improved financial infrastructure. Robinson (1979) contended that economic growth primarily drives financial development, reversing the cause-and-effect relationship. The financial sector's pivotal role in economic growth, as articulated in Lucas's (1988) stylized statement, stands in contrast to the findings of Modigliani and Miller (1958), who,

assuming information symmetry and the absence of transaction costs, argued that the expansion of real sectors is unrelated to the growth of financial sectors. Some economists, such as Morck and Nakamura (1999), even suggest that banks may hinder rather than promote economic development.

Jahfer and Inoue (2014) found that both the development of the financial sector and economic growth are causal factors. Durusu-Ciftci et al. (2017), drawing data from 40 nations over the long term (1989 to 2011), attributed the observed rise in their study to a flourishing financial sector, advocating government support for the financial system. Arestis et al. (2015) discovered a robust positive correlation between financial growth and economic expansion.

Pradhan et al. (2018) found that growth and financial development are mutually causal across 35 countries from 1961 to 2015. Other studies have highlighted the indirect impact of financial development on growth. Yang (2019) demonstrated that financial development contributes to GDP growth in both high- and middle-income countries.

Bist (2018) explored the long-term relationship between financial development and economic growth using panel unit root and panel cointegration analysis in 16 low-income countries over two decades (1995 to 2014). Cross-sectional interdependence among the nations was evident, and the empirical findings from the long-term panel analysis revealed a robust and statistically significant relationship between financial development and economic growth.

Tran et al. (2020), examining over 40,000 Vietnamese enterprises, assessed the impact of local financial development on firm growth, identifying corruption as a major impediment. Mengesha and Berde (2023) examined the effect of improvements in a country's financial infrastructure on GDP growth from 1980 to 2021, uncovering a reverse causality linking economic growth to financial sector development.

Economic growth, as emphasized by Barro (1991), is closely linked to human capital improvement, reduced government spending, and macroeconomic stability. King and Levine (1993) expanded upon Barro's framework to include financial indicators. Levine (1997) introduced the standard indicator of financial depth, defined as the ratio of total liquid liabilities of the financial system to GDP. Financial development positively affecting economic growth led to financial depth becoming the standard measure for studying the interaction between finance and economic growth.

Ang (2008) studied how progress in the banking sector influenced GDP expansion in Malaysia from 1960 to 2003, showing a long-term positive effect of financial development on economic growth. Nguyen et al. (2019) employed the generalized method of moments (GMM) technique, highlighting the positive impact of stock and bond markets on economic growth in middle-income economies. Pradhan et al. (2017), analyzing GDP growth and four financial development measures from 1991 to 2011, identified unidirectional and bidirectional causation between the variables, advocating increased access to investment capital and stock market development.

Nevertheless, there is growing evidence challenging the conventional belief that deeper financial markets invariably lead to faster economic growth. Klein and Olivei (2008) concluded that capital flow liberalization's beneficial effects are primarily seen in industrialized economies. However, Caporale et al. (2015) and Stolbov (2017) discovered that a causal relationship between financial depth and economic growth is not universal.

Polemis et al. (2020) found no robust or linear influence of conventional measures of financial depth, such as broad money and domestic credit to GDP ratios, on economic development. Isiaka et al. (2021) observed a detrimental impact of financial depth on economic growth, irrespective of the measurement parameter.

Alfaro et al. (2004) emphasized the vital role of finance in facilitating foreign direct investment (FDI) contributing to economic growth. Kutan et al. (2017) focused on the roles of FDI and institutional quality in MENA nations, revealing that financial development benefits these countries, subsequently boosting economic growth.

Using the VECM method, Biplob and Halder (2018) found a one-way relationship between financial loans and growth when examining the connection between capital flow

liberalization, financial depth, and economic growth in Bangladesh. They emphasized the importance of a robust financial sector for overall economic growth, particularly the role of private sector credit and domestic investment.

Mohanty and Bhanumurthy (2019) and Aziz et al. (2023) identified a strong correlation between financial development and economic growth, along with a bidirectional causal link between the two. Furthermore, Mohanty and Bhanumurthy identified finance as the leading predictor of the Indian economy, emphasizing private savings and investment as key factors influencing development. Anwar and Nguyen (2011) employed a panel GMM model, analyzing data from 61 Vietnamese provinces between 1997 and 2006, revealing a causal link between improved access to capital and economic development, where both a large money supply and gross domestic savings played pivotal roles.

Shahbaz et al. (2013) conducted a multivariate framework analysis, exploring the dynamic relationships between economic expansion, energy consumption, financial advancement, and international trade. Their findings indicated long-term interconnections among these variables, using ARDL bounds testing, and highlighted a two-way connection between financial development and economic growth. Additionally, Shahbaz et al. (2015) found that the financial sector positively impacted economic expansion, with trade openness also fostering economic development.

Bist and Bista (2018), using the ARDL model over a 30-year period (1984 to 2014), identified a strong positive unidirectional relationship between economic growth and financial stability in Nepal, while observing a negative correlation between growth and both trade openness and gross domestic credit. Furthermore, Rahman et al. (2020), studying Pakistan using the Markov switching model and data from 1980 to 2018, affirmed the role of finance in fostering development across high- and low-growth nations, with high-income regions experiencing more rapid expansion. Government expenditure and trade liberalization were identified as contributors to economic growth.

Zhang and Zhou (2021), explore various theoretical schools of thought and empirical discoveries on this nexus, with the goal of developing a cohesive, microfounded model in a small open-market scenario to accommodate multiple theoretical possibilities and actual data. The model is then adjusted to reflect some well-documented stylized facts. Numerical models reveal that in the long term, the welfare-maximizing level of financial development is lower than its growth-maximizing level. In the near run, the price channel (via the global interest rate) outweighs the quantity channel (via financial productivity), highlighting the critical importance of international collaboration in addressing systemic risk.

Despite the ongoing debate among academics regarding the causative link between financial development and economic growth, this study seeks to contribute to the literature by testing the hypothesis of a causal relationship between financial development and economic growth across a broader sample of MENA countries. The primary objectives of this research are (a) to determine the association between financial development and economic growth and (b) to quantify the existence and direction of causality between financial development indicators and economic growth in the MENA region.

## 3. Methodology

In accordance with recent advancements in second-generation panel unit-root testing, exemplified by Bai and Ng (2004), Bist (2018), Dumitrescu and Hurlin (2012), Moon and Perron (2004), and Pesaran (2007), there is a pressing need for innovative panel non-causality tests that explicitly account for various forms of dependencies among panel members. To address this need, this paper aligns with the methodological approach proposed by Bist (2018), Mengesha and Berde (2023), and Pradhan et al. (2017) in developing a log–log linear regression model. This model is designed to investigate the finance–growth nexus while considering the presence of other covariates.

Recent literature on the finance–growth nexus underscores the significance of examining this relationship through the lens of endogenous growth theory, as advocated by Bist and Bista (2018), Guru and Yadav (2019), Haque et al. (2022), Mengesha and Berde



(2023), and Pradhan et al. (2017). In our model, the endogenous variable is economic growth, calculated as the natural logarithm of a nation's GDP per capita at time t. Per capita income, determined by dividing a country's gross domestic product by its population (Beylik et al. 2022), serves as a crucial economic metric for assessing a country's level of development. While acknowledging that economic development depends on a multitude of factors (Aye and Edoja 2017), our analysis focuses solely on factors for which empirical data are readily available.

The primary variable of interest in our investigation is the degree of financial development (FD), represented using a proxy variable due to its indirect measurability. To enhance accuracy, we adopt a comprehensive measure of financial development developed by Svirydzenka (2016), moving away from earlier studies that relied solely on domestic lending to the private sector as a share of GDP as a proxy for financial development. This encompassing index, recognized as the IMF financial development index, assesses financial depth, access, and efficiency. Recent empirical research strongly supports its superiority as a measure of financial development (Chen et al. 2020; Mengesha and Berde 2023; Raifu et al. 2023).

Trade openness, quantified as the ratio of total trade value to gross domestic product, is posited to contribute to an increase in the technology index and subsequently stimulate economic growth (Jalil and Rauf 2021). This linkage can be attributed to trade's facilitation of technology diffusion, information and skills transfer, leading to more efficient resource utilization and increased factor productivity, all of which support a nation's economic growth (Islam et al. 2022; Mtar and Belazreg 2023).

Furthermore, the foundations of economic development rest on capital and labor. Investment has a positive impact on economic growth, as indicated by both the Cobb Douglas production function and other models (Bist 2018; Narayan and Narayan 2013). Additionally, a nation's overall development hinges on its labor force (Bist 2018).

Conversely, inflation exerts an influence on both economic growth and a nation's financial operations by altering interest rates, which directly affect the activities of financial institutions, including deposit mobilization and lending (Beck et al. 2000; Bist 2018; Christopoulos and Tsionas 2004; Levine et al. 2000). These variables feature prominently in the literature for regulating the connection between finance and growth.

Therefore, the primary aim of this study is to provide empirical evidence regarding the relationship between financial sector growth and economic expansion in MENA region countries, a task undertaken via our log–log regression model. After a comprehensive review of the literature, we have selected investment, trade openness, inflation, and the labor force as control variables. As a result, the model takes the following form:

$$\text{LGDPPC}_{it} = \beta 0_i + \beta 1i\text{LFDI}_{it} + \beta 2i\text{LINV}_{it} + \beta 3i\text{LOPE}_{it} + \beta 4i\text{LINF}_{it} + \beta 5i\text{LLF}_{it} + \mu_{it}$$

We use the natural log of these variables in our estimation.

(LGDPPC): Economic growth, defined as the natural log of gross domestic product per capita, measured in constant US dollars using the purchasing power parity approach and 2017 international dollars. This variable serves as the dependent variable.
(LFDI): A proxy for the development of the financial sector, represented as the natural log of the IMF Financial Development Index.
(LINV): The natural log of the ratio of total investment to GDP.
(LOPE): Trade openness, as the natural log of the import plus export to GDP ratio.
(LINF): The natural log of the inflation rate.
(LLF): The natural log of the labor force as a percentage of the total population.
($\mu_{it}$): The error term in our model.

The model accounts for the heterogeneity among the various nations by allowing for distinct intercepts and slope coefficients for each country. In this context, the country-specific fixed effect is denoted as $\beta 0i$, while the long-run coefficients for private credit, investment, trade openness, consumer price index, and labor force are represented by $\beta 1i$, $\beta 2i$, $\beta 3i$, $\beta 4i$, and $\beta 5i$, respectively.

Data for this model are sourced from the World Economic Outlook publication published by the World Bank and the Penn World Table (PWT 10.01) for the period spanning the years from 1980 to 2021. Consequently, due to data availability constraints, our study is limited to 16 nations. The countries included in our analysis are Algeria, Bahrain, Egypt, the Islamic Republic of Iran, Israel, Jordan, Kuwait, Lebanon, Libya, Morocco, Oman, Qatar, Saudi Arabia, Tunisia, Turkey, and the United Arab Emirates.

## 4. Empirical Analysis

### 4.1. Descriptive Analysis

Table 1 shows the descriptive statistics for the overall sample of 16 nations in the panel. The logarithmic values of LGDPPC, LFDI, LINV, LOPE, LINF and LLF were used.

**Table 1.** Descriptive Statistics for Countries in the MENA Region.

| Descriptive Statistics | | | | | | |
|---|---|---|---|---|---|---|
| Countries | LGDPPC | LFDI | LINV | LOPE | LINF | LLF |
| Algeria | 9.2 | −2.09 | 3.53 | 0.72 | 2.14 | −1.52 |
| Bahrain | 10.75 | −0.99 | 3.2 | 0.16 | 0.51 | −0.90 |
| Egypt | 9.04 | −1.26 | 3.05 | −0.30 | 2.46 | −1.39 |
| Iran | 9.32 | −1.21 | 3.59 | 0.44 | 3.03 | −1.35 |
| Israel | 10.29 | −0.71 | 3.17 | −2.81 | 3.66 | −0.84 |
| Jordan | 9.19 | −0.73 | 3.29 | 1.62 | 1.49 | −1.50 |
| Kuwait | 10.77 | −1.04 | 2.95 | 2.94 | 1.18 | −0.73 |
| Lebanon | 9.66 | −1.36 | 3.17 | −2.95 | 3.50 | −1.37 |
| Libya | 10.57 | −2.09 | 4.40 | 1.25 | 1.86 | 3.32 |
| Morocco | 8.55 | −1.47 | 3.35 | −1.26 | 1.29 | −1.14 |
| Oman | 10.25 | −1.14 | 3.10 | 0.28 | 0.63 | −0.99 |
| Qatar | 11.41 | −0.76 | 3.66 | 3.05 | 1.19 | −0.54 |
| Saudi Arabia | 10.76 | −0.99 | 3.15 | 1.25 | 0.21 | −1.14 |
| Tunisia | 8.93 | −1.61 | 3.17 | −1.74 | 1.69 | −1.2 |
| Turkey | 9.75 | −1.04 | 3.23 | 0.8 | 3.67 | −1.13 |
| UAE | 11.45 | −1.17 | 3.2 | 2.29 | 1.32 | −0.59 |
| Average | 9.993 | −1.229 | 3.326 | 0.359 | 1.864 | −0.813 |
| Standard Deviation | 0.909 | 0.420 | 0.346 | 1.819 | 1.118 | 1.144 |

(Source: Authors' own calculations).

This table presents a 41-year average of the six variables used in this research. For the dependent variable LGDPPC, it seems that the oil-rich Arab countries, Bahrain, and Israel have the highest average GDP per capita, the highest ever being the UAE with 11.45, then Qatar with 11.41. The lowest ever is Morocco with 8.55, then Tunisia with 8.93. These results are applicable to the outcomes of huge boom of global oil prices during the era of study. For the independent variables, it was found that two variables have negative signs; LFDI has negative coefficients for all countries and LIF has negative coefficients for all countries except for Libya. On the other hand, the other three explanatory variables have positive coefficients for the majority of countries; LINV and LINF have positive coefficients for all nations. These results are applicable to the economic theory since GDP per capita is positively impacted by investment, inflation, and trade openness. The countries which benefited most from investment are Libya with 4.4, then Qatar with 3.66, Iran with 3.59, and Algeria with 3.53; the lowest is Kuwait with 2.95, then Egypt with 3.05. The countries which benefited most from trade openness are Qatar with 3.05 then Kuwait with 2.94 and UAE with 2.29; where the lowest are Lebanon with −2.95, then Israel with −2.81 and Tunisia with −1.74. These results are applicable to the situation of these countries' tendency towards more openness, in the case of Arab Gulf countries, and more restrictions in countries such as Lebanon and Tunisia. For inflation, it is noted that countries with high and persistent inflation rates, such as Turkey, Irael, Lebanon, Iran, and Egypt, have the highest positive impact of inflation on GDP per capita. On the other hand, this positive

impact of inflation reached its minimum in countries having low levels of inflation, such as Saudi Arabia, Bahrain, and Oman, as a result of being rich nations.

### 4.2. Measuring Financial Development

Numerous measures have been proposed in the literature to assess the development of a country's financial sector. Initially, these measures primarily included monetary aggregates such as M1 and M2. However, these metrics are more indicative of the financial system's capacity to provide transaction services rather than its ability to facilitate the transfer of funds from savers to borrowers (Hashmi and Bhatti 2019). Similarly, commonly used variables in the literature encompass credit to the private sector (Beck et al. 2000; Levine et al. 2000), liquid liabilities (King and Levine 1993), and deposit liabilities (Christopoulos and Tsionas 2004). Additionally, stock market indicators have been employed by various researchers as proxies for measuring financial development.

More recently, the International Monetary Fund (IMF) has introduced a set of indicators (Svirydzenka 2016) to assess a nation's financial development comprehensively. These indicators collectively constitute the Financial Development Index (FDI), comprising the Financial Institutions Efficiency [FIE] Index, Financial Institutions Depth [FID] Index, and Financial Institutions Access [FIA] Index, which evaluate the accessibility, depth, and efficiency of financial institutions. Furthermore, the Financial Markets Efficiency [FMEI] Index, Financial Markets Depth [FMDI] Index, and Financial Markets Access Index [FMAI] are three financial market indices that employ similar metrics to assess the state and pace of development in financial markets. Consequently, this paper employs the IMF's Financial Development Index (FDI) to conduct econometric analyses for the MENA region countries under study.

To account for the finance–growth nexus, this paper incorporates macroeconomic variables, including trade openness (measured as the sum of imports and exports as a percentage of GDP), investment as a percentage of GDP, labor force (defined as the proportion of the economically active population aged 15 and older to the total population), and inflation (measured as the consumer price index). The ratio of GDP from imports plus exports, reflecting trade openness, provides insight into a country's economic status, as trade connects nations to technological advancements achieved by their trading partners.

Furthermore, as argued by Yanikkaya (2003), trade offers developing nations access to crucial investment and intermediate goods essential for their developmental processes. In a manner akin to its influence on growth, inflation also affects a nation's financial activities by altering interest rates, thereby directly impacting the operations of banking and financial institutions, including deposit collection and mobilization. Alongside these factors, capital and labor are fundamental pillars in any theory of economic development. In various models, capital stock is shown to have a favorable impact on economic growth (Narayan and Narayan 2013). Similarly, a nation's overall development hinges on its labor force.

These variables have been extensively utilized in the literature to explore the finance–growth nexus, as evidenced by studies such as those conducted by Beck et al. (2000), Christopoulos and Tsionas (2004), Levine et al. (2000), Menyah et al. (2014), Narayan and Narayan (2013), Salahuddin and Gow (2016), and Samargandi et al. (2014).

### 4.3. Data Analysis Procedures

Our data analysis encompasses four essential steps. First, it involves determining the integration levels of the variables. Second, it examines whether these variables exhibit long-term co-integration. Third, it entails estimating the parameters associated with long-term co-integration. Finally, the fourth step involves testing the short-term causal relationship between financial development and economic growth.

### 4.4. Integration Levels

In heterogeneous panel data analysis, Im et al. (2003) and Maddala and Wu (1999) panel unit root tests are the most commonly employed approaches, as indicated in the literature. While these tests allow for individual unit root processes within a panel, they do not address the issue of cross-sectional dependence (Pesaran 2007). Therefore, before conducting first-generation unit root tests akin to those conducted by Im et al. (2003) and Maddala and Wu (1999), it is imperative to assess cross-sectional dependence. To tackle this concern, the study employs the Cross-Sectional Augmented IPS (CIPS) test, a second-generation panel unit root test developed by Pesaran (2007). The results of the second-generation panel unit root (CIPS) test and the investigation of cross-sectional dependence in the series are presented in Table 2.

**Table 2.** Results of second-generation panel unit root (CIPS) and cross-sectional dependence Pesaran CD test.

| | Variables | | | | | |
|---|---|---|---|---|---|---|
| **Tests** | **LGDPPC** | **LFDI** | **LINV** | **LOPE** | **LINF** | **LLF** |
| Pesaran CD | 14.51 * | 23.7 * | 7.63 * | 16.32 * | 15.39 * | 50.18 * |
| *p*-value | 0.000 | 0.000 | 0.001 | 0.002 | 0.000 | 0.003 |
| CIPS Level | −2.903 | −2.29 | 1.258 | 0.16 | 0.51 | −0.90 |
| *p*-value | 0.213 | 0.146 | 0.251 | 0.09 | 0.149 | 0.111 |
| CIPS (First Difference) | −3.637 * | −2.430 * | −2.517 * | −2.043 * | −2.956 * | −2.045 * |
| *p*-value | 0.001 | 0.000 | 0.003 | 0.005 | 0.000 | 0.000 |

(Source: Authors' own calculations). * Indicates significance at 1 percent.

To evaluate cross-sectional dependence for all variables, Pesaran's CD test is used where the following hypothesis test is applied:

**H0:** *There is no cross-sectional dependence.*

**H1:** *There is cross-sectional dependence.*

For all variables, the Pesaran CD test results yield *p*-values below 0.05, leading to the rejection of the null hypothesis. This indicates the presence of cross-sectional dependence within the dataset. Consequently, it becomes imperative to apply a second-generation panel unit root test to address the limitations associated with cross-sectional dependence, thus ensuring more accurate results compared to the first-generation unit root tests when cross-sectional dependence among variables is present.

The second-generation panel unit root test is formulated as follows:

**H0:** *The series is not stationary (Unit Root Test is Present).*

**H1:** *The series is stationary.*

Upon reviewing the second row of Table 2, it is evident that all variables are non-stationary at the level, as the null hypothesis is not rejected. Subsequently, the CIPS test is conducted, which results in the rejection of the null hypothesis, with *p*-values consistently below 0.05 for all variables in the series. This signifies that the variables under examination are stationary at the first difference. Consequently, the investigation confirms that these variables are integrated at order one, denoted as 'I(1) variables'.

### 4.5. Co-Integration and Long-Run Relationship Estimation

Having established that the variables are integrated at order one I(1), the next step involves conducting a co-integration test among these variables. In accordance with Pedroni (2004), this paper employs a panel co-integration test, which computes seven test statistics, as detailed

in Table 3. The Pedroni test comprises two sets of tests: the first set comprises panel co-integration statistics within the dimension and includes four statistics—namely, the panel v-statistic, panel rho-statistic, panel PP-statistic (nonparametric), and panel ADF-statistic (parametric). The second set encompasses three statistics, referred to as between-dimension statistics or group mean panel co-integrating statistics, and comprises the group rho-statistic, group PP-statistic (nonparametric), and group ADF-statistic (parametric). The null hypothesis for this comprehensive test posits that there is no co-integration within the series, and this hypothesis is scrutinized using the seven different co-integration test statistics. The results of these seven statistics are presented in Table 3.

**Table 3.** Results of Pedroni panel co-integration test.

| Tests | Statistics | Probability | Weighted Statistics | Probability |
|-------|-----------|-------------|---------------------|-------------|
| Alternative hypothesis: common AR coefficients (within-dimension) | | | | |
| Panel v-Statistic | −0.563 * | 0.0071 | −2.102460 * | 0.0098 |
| Panel rho-Statistic | 0.892 * | 0.0081 | 0.581188 | 0.7194 |
| Panel PP-Statistic | −1.059 * | 0.0146 | −2.350003 * | 0.0094 |
| Panel ADF-Statistic | −0.976 * | 0.0164 | −2.601474 * | 0.0046 |
| | Statistic | | | |
| Alternative hypothesis: individual AR coefficients (between-dimension) | | | | |
| Group rho-Statistic | 1.808 | 0.9647 | | |
| Group PP-Statistic | −1.589 ** | 0.0560 | | |
| Group ADF-Statistic | −0.569 | 0.0128 | | |

\* Indicates significance at 1 percent. \*\* Indicates significance at 5 percent. (Source: Authors' own calculations).

The results presented in Table 3 reveal that four out of the seven statistics are significant at the 1 percent level, with an additional statistic significant at the 5 percent level. However, the remaining two statistics do not achieve significance. Consequently, the null hypothesis, positing no co-integration within the series, is rejected. It can therefore be concluded that the variables comprising GDP per capita growth rate, financial development, investment, trade openness, inflation rate, and labor force share a long-run equilibrium relationship.

Subsequently, the next step involves estimating the associated long-run co-integration parameters for this set of variables. It is imperative to employ panel data analysis techniques that address endogeneity, heterogeneity among the variables and nations, as well as integration and co-integration aspects of the data. In this regard, this study utilizes the Fully Modified Ordinary Least Square (FMOLS) and Dynamic Least Square (DOLS) methods to assess the long-run connection between the co-integrated variables in the panel.

Christopoulos and Tsionas (2004) put forth three compelling arguments in favor of FMOLS application in a co-integrated panel. They contend that FMOLS allows for consistency between the long-run relation and short-run adjustments, effectively handles the endogeneity of regressors, and respects the time-series properties of the data by explicitly considering integration and co-integration properties.

Similarly, DOLS rectifies errors by incorporating leads, lags, and contemporaneous values of the regressors into the static regression (Kao and Chiang 2001). Consequently, this study employs both FMOLS and DOLS approaches to estimate the long-run parameters, ensuring the robustness of the results. After calculating the panel estimates of long-run parameters, FMOLS is further employed to estimate the long-run values across the countries, enhancing the robustness of the findings.

### 4.6. Estimating the Long-Run Co-Integrating Parameters

After estimating the long-run parameters, the direction of causation between the variables is examined. To analyze the causation nexus while accounting for the panel of 16 low-income countries and the presence of cross-sectional dependence, this study employs a pairwise panel causality test developed by Dumitrescu and Hurlin (2012). Importantly,

this technique yields reliable standardized panel statistics for small samples, even in the presence of cross-sectional dependence (Dumitrescu and Hurlin 2012).

The test statistic operates on the premise that all coefficients differ across the cross-sections because it relies on the individual Wald statistics of Granger non-causality averaged across the cross-sectional units. It is worth noting that, for this test, the variables must be stationary at the level since the test is conducted on the initial variation in the series. Consequently, the findings are deemed to reflect short-term causal relationships among the variables.

Table 4 reveals that the results obtained from the FMOLS and DOLS methods are highly consistent concerning sign, value, significance, and magnitude for each variable. The exception lies in the coefficient of financial development, where the signs differ.

**Table 4.** Results of fully modified OLS and dynamic OLS techniques.

| Variables | FMOLS | | | DOLS | | |
|---|---|---|---|---|---|---|
| | Coefficient | T-Statistic | *p*-Value | Coefficient | T-Statistic | *p*-Value |
| LFDI | 0.015 ** | 0.0740 | 0.043 | −0.215 *** | −0.5453 | 0.0906 |
| LINV | 1.93 * | 20.09 | 0.002 | 1.856 * | 6.9144 | 0.0000 |
| LOPE | −0.809 * | −3.6929 | 0.000 | −0.432 | −0.9148 | 0.956 |
| LINF | −0.041 *** | −1.163 | 0.091 | −0.095 | −1.1237 | 0.875 |
| LLF | −3.592 * | −9.681 | 0.000 | −3.665 ** | −4.1260 | 0.036 |
| R-squared | 0.91 | | | 0.90 | | |

(Source: Authors' own calculations). * Indicates significance at 1 percent. ** Indicates significance at 5 percent. *** Indicates significance at 10 percent.

The findings indicate that a 1% increase in the financial development index leads to a 0.015% increase in per capita income, significant at the 5% level. Furthermore, a 1% increase in investment as a percentage of GDP results in a substantial 1.9% increase in per capita income, significant at the 1% level. In contrast, three coefficients exhibit a negative relationship with per capita income growth. Specifically, a 1% increase in the openness of the economy reduces per capita income by 0.809%, significant at the 1% level. Additionally, a 1% increase in the inflation rate in the economy leads to a 0.041% decline in per capita income, significant at the 10% level. Finally, a 1% increase in the labor force in the economy results in a substantial 3.59% decrease in per capita income, significant at the 1% level.

The presence of several insignificant coefficients can be attributed to significant random variation, which may obscure the existence of a clear significant effect, especially in cases where data reliability is compromised in some countries within our dataset. Additionally, the possibility of multicollinearity in the data, wherein independent variables are correlated, has been considered. However, this study rejects the notion of multicollinearity, as it is a well-established model in the literature, and prior studies have not identified this issue in the endogenous growth model.

Since this paper has estimated the long-run panel coefficients, the next step involves forecasting the long-run estimates of the co-integrating relationship for each individual country. This step is crucial for understanding the impact of financial development on GDP per capita growth in the specified MENA region group of countries. The FMOLS model has been employed to make these long-run coefficient predictions, and the results of these forecasts are presented in Table 5.

Table 5 illustrates a predominantly positive relationship between financial development and GDP per capita growth in approximately 9 out of 16 countries in this group. These countries include Algeria, Bahrain, Egypt, Iran, Lebanon, Morocco, Oman, Tunisia, and Turkey. However, this relationship is statistically significant in only 8 countries within the series.

Additionally, a positive correlation between trade openness and GDP per capita growth is observed in 11 out of the 16 countries, encompassing Algeria, Iran, Israel, Jordan,

Libya, Morocco, Oman, Qatar, Saudi Arabia, Tunisia, and Turkey. However, the relationship reaches statistical significance in only 7 countries in the series.

**Table 5.** Long-run coefficients using FMOLS Models for 16 MENA countries (dependent variable: LGDPPC).

| | Independent Variables | | | | | | |
|---|---|---|---|---|---|---|---|
| Countries | Constant | LFDI | LINV | LOPE | LINF | LLF | $R^2$-Adjusted |
| Algeria | 10.137 | 0.61 | 0.21 | 0.080 | −0.03 | 0.20 | 0.854 |
| Bahrain | 10.735 | 0.132 | 0.039 | −0.047 | 0.0004 | −0.052 | 0.231 |
| Egypt | 13.456 | 0.329 | −0.375 | −0.139 | 0.064 | 2.270 | 0.883 |
| Iran | 10.436 * | 0.312 * | 0.113 | 0.082 | −0.031 | 0.727 * | 0.787 |
| Israel | 12.405 * | −0.038 | −0.066 * | 0.176 ** | −0.034 | 2.179 * | 0.980 |
| Jordan | 8.556 * | −0.403 | 0.103 | 0.273 | −0.022 | 0.01 | 0.084 |
| Kuwait | 12.048 * | −0.330 | −0.245 *** | −0.537 | −0.001 | 1.330 ** | 0.422 |
| Lebanon | 8.469 * | 0.962 * | 0.549 * | −0.071 | 0.010 | −0.514 | 0.197 |
| Libya | −0.622 | −0.841 | −0.419 * | 1.368 * | −0.031 | −11.591 * | 0.763 |
| Morocco | 10.244 * | 0.524 * | −0.187 | 0.624 *** | −0.031 | −0.040 | 0.902 |
| Oman | 12.051 * | 0.759 * | −0.258 *** | 0.172 | −0.004 | 0.130 | 0.557 |
| Qatar | 12.211 * | −1.063 ** | −0.356 | 2.005 * | −0.021 | 0.269 | 0.116 |
| Saudi Arabia | 10.517 * | −0.548 * | 0.201 *** | 0.350 * | 0.009 | 0.745 * | 0.738 |
| Tunisia | 13.796 * | 0.420 * | 0.062 | 0.066 | 0.032 | 3.718 * | 0.934 |
| Turkey | 10.459 * | 0.201 * | 0.561 * | 0.435 * | −0.115 * | 1.407 * | 0.944 |
| UAE | 12.692 * | −0.229 | −0.327 *** | −0.593 ** | 0.060 *** | 0.936 | 0.758 |

\* Indicates significance at 1 percent. \*\* Indicates significance at 5 percent. \*\*\* Indicates significance at 10 percent. (Source: Authors' own calculations).

Similarly, there is a positive association between the labor force as a percentage of the population and GDP per capita growth in 12 out of the 16 countries, including Algeria, Egypt, Iran, Israel, Jordan, Kuwait, Oman, Qatar, Saudi Arabia, Tunisia, Turkey, and the UAE. Nevertheless, the relationship is statistically significant in only 7 countries within the series.

In contrast, a negative relationship between inflation and GDP per capita growth is evident in 10 out of the 16 countries, involving Algeria, Iran, Israel, Jordan, Kuwait, Libya, Morocco, Oman, Qatar, and Turkey. However, statistical significance is observed in only two countries: Turkey and the UAE.

Regarding the variable LINV, it is noteworthy that half of the countries exhibit a positive relationship between investment as a percentage of GDP and GDP per capita growth, while the other half show a negative relationship. Additionally, the significance of this relationship varies, with half of the countries having non-significant results.

It is essential to acknowledge that despite a high R-squared value, a significant proportion of the estimated coefficients appear to be statistically insignificant. The prevalence of insignificant coefficients suggests that this issue may be attributed to a high level of random variation and underscores the need for more accurate and reliable data in our dataset.

## 5. Discussion

The finance–growth nexus has long been a subject of extensive debate in the realm of financial economics, with two primary schools of thought. Schumpeter and his followers initiated the first school, asserting that 'financial development is indispensable for economic growth'. According to this perspective, financial progress stimulates economic growth by influencing savings, investment, and technological innovations.

In contrast, neo-classical economists, led by Lucas (1988), argue that 'finance is not a primary source of growth.' In light of this ongoing debate, our study aims to investigate the finance–growth nexus in 16 selected MENA countries. While substantial literature exists on this topic, particularly concerning cross-sectional and time series econometric approaches,

significant gaps persist in terms of methodology, data properties, and econometric analysis specific to MENA region countries.

Historically, investigations into the relationship between finance and growth have relied on cross-sectional data or time series econometric methods. However, the scarcity of extensive time series data presents a challenge for time series approaches (Agbetsiafa 2004; Atindéhou et al. 2005; Ghirmay 2004; Odhiambo 2005, 2007). Additionally, panel data studies have faced criticism, particularly those combining data from low-income, middle-income, or high-income countries due to substantial variations across these groups. Furthermore, while previous research has employed first-generation unit root tests, sensitive to cross-sectional dependence (Pesaran 2007), the methods used to estimate long-run co-integrating equations have come under scrutiny, as they do not adequately consider integration and co-integration phenomena.

The contributions of our study to the existing literature are noteworthy. Firstly, we have aggregated data from 16 MENA nations, spanning a substantial time frame from 1980 to 2021, allowing for a comprehensive examination of the finance–growth nexus. Secondly, we have implemented second-generation panel unit root tests to account for cross-sectional dependence, a critical consideration often overlooked in earlier research. Lastly, we have estimated co-integrating vectors using Fully Modified Ordinary Least Squares (FMOLS) and Dynamic OLS (DOLS) methods, which explicitly consider integration and co-integration properties in the analysis.

Analyzing the outcomes of the long-run coefficients, we find that increased financial development generally corresponds to greater economic growth and per capita income growth in the majority of MENA nations, supporting the first school of thought. However, exceptions, including Israel, Jordan, Kuwait, Libya, Qatar, Saudi Arabia, and the UAE, demonstrate a contrasting relationship. The outcomes of Ductor and Grechyna (2015), Grassa and Gazdar (2014), Mhadhbi (2014), and Narayan and Narayan (2013) are comparable to this result.

Moreover, our findings align with the idea, which has been investigated by Ductor and Grechyna (2015), that an excessive increase in financial development, not accompanied by a corresponding increase in real production, may lead to a negative impact on GDP growth. This suggests that there is an optimal level of financial development influenced by an economy's unique characteristics and production capacities.

Furthermore, increased trade openness is associated with higher economic growth and GDP per capita income growth in most of the examined MENA nations, indicating their ability to produce goods and services domestically and export them. Exceptions such as Bahrain, Egypt, Kuwait, Lebanon, and the UAE can be attributed to weaker performance in productive institutions and the export sector.

Labor force emerges as a significant driver of economic growth and GDP per capita income across these countries, in line with economic theory, which suggests that increased labor, along with technological progress, leads to greater productivity and economic growth. This finding corresponds with the research of Kamala and AboElsoud (2023).

Investment presents a more complex picture, as 8 out of 16 countries show a negative relationship between investment and GDP per capita income growth. This is consistent with studies suggesting that accelerated financial development can misallocate resources, resulting in reduced growth (Bolton et al. 2011; Philippon 2010; Santomero and Seater 2000). Overemphasis on the financial sector can lead to resource misallocation, stagnation in other sectors, and hindered short- and long-term growth rates.

Lastly, our analysis suggests that inflation is negatively related to GDP per capita income growth within the context of progressive financial development. Financial innovation and liberalization, integral aspects of financial development, contribute to increased systemic risk. Elevated systemic risk, in turn, leads to more frequent and severe crises, ultimately resulting in higher inflation and lower economic growth rates (Allen and Gale 2004; Allen and Carletti 2006; Gennaioli et al. 2012; Wagner 2007).

Regarding statistically insignificant coefficients, it is essential to acknowledge that while random variation can sometimes obscure significance, it does not inherently negate the existence of an effect. Careful consideration is required to interpret these findings, and they may be attributed to data limitations or, in some cases, potential multicollinearity among independent variables. Nevertheless, our model provides valuable insights into the complex interplay between financial development and economic growth in MENA nations, contributing to a deeper understanding of this critical relationship.

Needless to say, our findings can be extended to regions other than the MENA region. As a result, the Pedroni's panel cointegration estimates shown in our analysis give crucial support for the hypothesis that a long-run cointegrating relationship between financial development and economic growth appears in low- and middle-income countries, such as African and Latin American nations. We found that in regions such as African and Latin American nations, the flow of credit to the private sector is very low. Therefore, one of the important policy implications of this paper finding is that policymakers have to place more emphasis on the policies that provide a favorable environment for the private sector to grow.

## 6. Conclusions

Using cross-sectional and time series data spanning the years 1980 to 2021, this study investigates the long-term relationship between finance and economic growth in 16 MENA region economies. Importantly, this research incorporates both cross-sectional and time series data, addressing cross-sectional dependence, a dimension often overlooked in previous studies. Employing the second-generation panel unit root test, we estimated long-run parameters via Fully Modified OLS (FMOLS) and Dynamic OLS (DOLS) methods, revealing a significant long-term co-integrating relationship between financial development and economic growth across these 16 MENA nations.

Our analysis, based on the results of the Pedroni cointegration test, establishes a persistent link between GDP per capita income growth, financial development, trade openness, investment, inflation rate, and the labor force throughout the study period. Notably, our findings underscore the profound and lasting influence of financial sector development on GDP per capita income growth. Among the individual countries studied, nine out of sixteen exhibit a substantial positive association between financial development and income growth, as per long-run estimates obtained using the FMOLS model. Conversely, seven countries demonstrate a negative relationship between finance and growth. Hence, our findings emphasize that the most favorable impact of financial development on growth is realized when it is in harmony with real sector growth. This beneficial effect diminishes when financial development outpaces real output growth significantly, possibly leading to adverse consequences.

Our conclusions remain robust across various financial progress metrics and are consistent in both cross-sectional and panel estimations. These findings align with theoretical studies that have previously explored the intricate interplay between financial sector and real sector technologies, influencing how the expansion of the financial sector affects GDP growth. These insights can guide the formulation of macro-prudential policies, urging policymakers in the MENA region to concentrate on policies that foster financial inclusion, encourage private sector engagement in the economy, mitigate the crowding-out effect in accessing finance, and bolster credit guarantees while reducing non-performing loans. It is also recommended to move towards financial policies in conjunction with a reduction in reliance on fiscal and monetary policies. Lastly, developing stock markets to facilitate increased access to investment capital should be a priority for MENA countries, fostering economic and income growth. Furthermore, enhancing the accuracy and reliability of economic data is essential, representing the foundational step toward shaping effective economic policies.

### 6.1. Policy Implications

The findings of this study offer significant policy insights for the MENA region. Firstly, policymakers in these economies should prioritize measures to promote financial inclusion. Enhancing the accessibility of financial services to a broader spectrum of the population can stimulate economic growth and reduce income disparities. Secondly, encouraging greater participation of the private sector in the economy is crucial. Policies that facilitate an environment conducive to private sector expansion, entrepreneurship, and innovation can be instrumental in fostering economic development. Thirdly, it is imperative to mitigate the crowding-out effect in accessing finance. This requires policymakers to streamline financial processes and reduce inefficiencies that may hinder access to financial resources for both businesses and individuals. Fourthly, the strengthening of credit guarantees should be considered as a policy option. Such measures can instill confidence among lenders, thereby increasing lending activity, which is vital for driving economic growth. Additionally, a focus on managing and reducing the percentage of non-performing loans is essential to maintain the stability and health of the financial sector. Lastly, it is recommended that policymakers concentrate on the development of stock markets as a means to increase access to investment capital. A robust stock market infrastructure can facilitate economic and income growth by attracting investment. Moreover, to maintain economic stability, it is suggested that policymakers reduce their reliance on traditional fiscal and monetary policies and incorporate financial policies into their macroeconomic strategies. These comprehensive policy recommendations are derived from the study's robust findings and can contribute significantly to fostering sustainable economic growth in the MENA region.

### 6.2. Limitations

Data Accuracy: Our findings are contingent on the quality and precision of available data. Efforts to enhance data accuracy and reliability are necessary.

Contextual Specificity: This study's applicability is confined to the MENA region, and generalizing findings to other regions may be limited. Diverse global contexts should be considered for a comprehensive understanding.

Model Sensitivity: Our results depend on model assumptions, which may vary with different specifications. Sensitivity analyses should be conducted to assess the stability of findings.

### 6.3. Future Research

Future research can expand upon this study by exploring several key areas. First, investigating the relationship between economic growth in specific nations and innovations in both the financial and real sectors is essential for a deeper understanding of their influence on economic development.

Second, examining the impact of financial liberalization after political regime changes can provide valuable insights into how political transitions affect the financial sector and, subsequently, economic growth.

Lastly, research can explore the potential for using technological advancements to predict economic recessions across various industries. However, this requires access to accurate and reliable data, underscoring the importance of data quality enhancement. These research directions can further enhance our comprehension of the interplay between finance, growth, and their contextual significance within the MENA region, ultimately offering valuable guidance for policymakers and stakeholders.

**Author Contributions:** Conceptualization, A.L.M.K. and K.A.; Formal analysis, K.A. and M.E.A.; Investigation, Software; A.L.M.K. and M.E.A.; Methodology, M.E.A., K.A. and A.L.M.K.; Validation, A.L.M.K., K.A. and M.E.A.; Writing—original draft; All authors have read and agreed to the published version of the manuscript.

**Funding:** The APC was funded partially by The British University in Egypt.

**Institutional Review Board Statement:** Not applicable.

**Informed Consent Statement:** This article does not contain any studies with human participants or animals performed by any of the authors.

**Data Availability Statement:** The data that support part of the findings of this study are available and freely accessed from World Bank, World Development Indicators available at https://databank.worldbank.org/source/world-development-indicators (accessed on 1 December 2022).

**Conflicts of Interest:** The authors declare no conflict of interest. In addition, the funders had no role in the design of the study; in the collection, analyses, or interpretation of data; in the writing of the manuscript, or in the decision to publish the results.

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
