# Peer review of "Modeling Finance–Growth Nexus in MENA Region Economies: A Panel Data Analysis"

_economies, doi:10.3390/economies11120290_

Round 1

Reviewer 1 Report

Comments and Suggestions for Authors

1-          Kindly interpret the descriptive statistics in Table 1 by conducting a comparative analysis.

2-          Kindly consider revising the title of Table 2 for improved clarity.

3-          Please disclose the name of the cross-sectional dependence test utilized.

4-          Kindly provide the critical values for the CIPS test in Table 2.

5-          Please present the p-values for the CD test statistics.

6-          You first employ a second-generational panel unit root test; however, employ a first-generational panel unit root test; please provide an explanation for this choice, or alternatively, consider using a second-generation panel cointegration test.

7-          Kindly conduct robustness tests to verify the accuracy of the findings.

Author Response

Manuscript Number: economies-2687255

Title: Modeling Finance-Growth Nexus in MENA Region Economies: A Panel Data Analysis

I would like to take this opportunity to convey my appreciation for allowing us to make revisions to our paper. I am sincerely thankful for the insightful and constructive comments that you have provided, as they have been instrumental in enhancing the quality and clarity of our work. The time and effort that you have invested in reviewing our paper are highly valued, and we are grateful for your continuous support and guidance during the publication process. I would like to reiterate our gratitude for the opportunity to revise our paper, and we are eager to receive your feedback on the revised version.

Reviewer 2 Report

Comments and Suggestions for Authors

The paper titled "Modeling Finance-Growth Nexus in MENA Region Economies: A Panel Data Analysis" aims to investigate the relationship between finance and economic growth in the Middle East and North Africa (MENA) region. The authors employ advanced quantitative approaches and the latest available data to address this challenge. The paper contributes in three significant ways: firstly, it utilizes a log-log linear regression model to examine the relationship between finance and economic growth while considering the influence of other factors for reliable results. Secondly, despite the growing significance of the MENA region in the global economic landscape, research on the relationship between finance and economic growth in this region remains limited. To address this gap, the authors employ a comprehensive dataset spanning from 1980 to 2021, comprising panel data from 16 nations. Finally, the authors utilize the broad-based financial development index developed by Svirydzenka (2016) from the International Monetary Fund (IMF) to investigate the relationship between finance and economic growth in the MENA region, enhancing the robustness of their empirical results.

There are, however, some points the authors can improve before being accepted.

  1. Lack of clarity in the introduction: The introduction of the paper could be more concise and clearer in presenting the research question and objectives. The authors could have provided a more detailed explanation of the significance of the research question and how it contributes to the existing literature.
  2. Limited scope: The paper focuses only on the relationship between finance and economic growth in the MENA region, which limits its generalizability to other regions. The authors could have included a discussion paragraph on other regions to provide a more comprehensive understanding of the finance-growth nexus.
  3. Insufficient discussion of limitations: The authors do not provide a comprehensive discussion of the limitations of their study. For example, the paper does not address the potential endogeneity problem that may arise from the relationship between finance and economic growth.
  4. Lack of policy recommendations: While the paper highlights the importance of financial development as a driver of economic prosperity, it does not provide specific policy recommendations for policymakers to support private sector growth and financial stability.
  5. Link with existing literature: There is a huge literature on the finance-growth nexus. A recent advance by Zhang & Zhou (2021) identify a nonlinear relationship and build that into their theoretical model. It would be good to include these new advancements in your literature review.

Reference:  

Zhang, Bo; Zhou, Peng. 2021. Financial development and economic growth in a microfounded small open economy model. North American Journal of Economics and Finance, 58, article number: 101544. DOI: 10.1016/j.najef.2021.101544.

Comments on the Quality of English Language

Overall, the quality of English language in the attached file appears to be satisfactory, but there are some repetitive language in some sections, which could have been avoided to improve the readability of the paper.

Author Response

(The authors gave the same response as above.)
